# Investigating the origin of subtelomeric and centromeric AT-rich elements in *Aspergillus flavus*

Arthur J. Lustig ORCID *

Department of Biochemistry and Molecular Biology, Tulane University Medical School, New Orleans, LA, United States of America

* alustig@tulane.edu

## Abstract

An in silico study of *Aspergillus flavus* genome stability uncovered significant variations in both coding and non-coding regions. The non-coding insertions uniformly consisted of AT-rich sequences that are evolutionarily maintained, albeit distributed at widely different sites in an array of *A. flavus* strains. A survey of $\geq$ 2kb AT-rich elements (AT $\geq$ 70%; ATEs) in non-centromeric regions uncovered two major categories of ATEs. The first category is composed of homologous insertions at ectopic, non-allelic sites that contain homology to transposable elements (TEs; Classes B, C, D, and E). Strains differed significantly in frequency, position, and TE type, but displayed a common enrichment in subtelomeric regions. The TEs were heavily mutated, with patterns consistent with the ancestral activity of repeat-induced point mutations (RIP). The second category consists of a conserved set of novel subtelomeric ATE repeats (Classes A, G, G, H, I and J) which lack discernible TEs and, unlike TEs, display a constant polarity relative to the telomere. Members of one of these classes are derivatives of a progenitor ATE that is predicted to have undergone extensive homologous recombination during evolution. A third category of ATEs consists of ~100 kb regions at each centromere. Centromeric ATEs and TE clusters within these centromeres display a high level of sequence identity between strains. These studies suggest that transposition and RIP are forces in the evolution of subtelomeric and centromeric structure and function.

## Introduction

*Aspergillus flavus* is an opportunistic pathogen in plants and humans. Its 37.5 Mb genome is separated into 8 chromosomes ranging in size from 6.5 Mb (chromosome 1) to 3.25 Mb (chromosome 8). The organization of *Aspergillus* genomes is unusual. Like many fungi, the genome is relatively compact containing approximately 13,500 genes in its genome [1] (on average, 1 gene per 3 kb of DNA). However, the genome is uniquely organized into gene clusters encoding proteins involved in the production of secondary metabolites, such as the toxin and carcinogen aflatoxin, as well as many medically and biologically important secondary metabolites [2–5]. While *A. flavus* gene organization has been well studied, the characteristics of genome

**Data Availability Statement:** All relevant data are within the paper and its Supporting Information Files.

**Funding:** AJL was funded in part by USDA NCA 58-6054-0-014. Arthur J. Lustig. The funders played no role in study design, data collection and

analysis, decision to publish, or preparation of the manuscript.

**Competing interests:** The authors have declared that no competing interests exist.

stability in *Aspergillus* are poorly understood. The presence of well-established and characterized strains of *A. flavus* makes it a unique vehicle for examining factors governing evolutionary challenges to genomic stability.

Transposable elements (TEs) are a significant force in genetic instability in prokaryotic and eukaryotic organisms. TEs in filamentous fungi include both Class 1 "copy and paste" LTR and non-LTR retrotransposition, and Class 2 "cut and paste" DNA-transposon classes. These transposable elements can also promote ectopic recombination [6]. In filamentous fungi, TEs contribute to non-allelic genomic rearrangement and genome restructuring. In *A. flavus*, active transposon identification has been limited to a single LINE element and a gypsy-related transposon [7–11]. Similarly, only a single functional Mariner class element, Crawler, has been found in *A. oryzae* [12], an ecotype of *A. flavus* [9]. A handful of additional active TEs have been identified in other *Aspergillus* species [13–16].

Additional intact and mutated TEs have been characterized based on comparisons between the *A. oryzae* and *A. nidulans* sequences [8]. Among TEs found in *A. oryzae* are sequences homologous to the DNA Mariner class of TEs [17–19], Gypsy class LTR-retrotransposons [20–22] and non-LTR retrotransposons [23]. LTR-retrotransposons have also been inferred initially by the presence of unique solo LTRs [11,24,25]. The Mariner and non-LTR transposons are highly mutated, inactive, and are likely to be evolutionary remnants of ancestral transposition, while Gypsy candidates retain significant coding capacity.

Repeat-induced point mutation (RIP) was initially identified as a C>T transition and cytosine methylation system during *Neurospora* pre-meiotic stages [26]. This process is responsible for the inactivation and silencing of fungal repeated elements, including transposons [27]. C to T transitions on either strand of the repeat lead to an elevated A+T bias within *Neurospora* centromeres and satellite sequences. Repeated transposition into the same genomic regions, possibly coupled with recombination among these regions, can result in an expansion of these A+T-rich regions [28]. Such regions are thought to be drivers of sequence divergence in most filamentous fungi [28–30].

Subtelomeric chromatin domains are often unique. Telomeres in many organisms, including the yeasts, nucleate the formation of densely packed and specialized nucleosomal regions that form heterochromatin [31]. Genes located within this region in *Schizosaccharomyces pombe* and *Saccharomyces cerevisiae*, as well as the filamentous fungi *Neurospora crassa* [32] and *Aspergillus nidulans* [33], are subjected to a form of transcriptional repression, termed telomere silencing. These position effects are controlled by a discrete set of proteins, the silencer information regulators, involved in histone modification and heterochromatic "spreading" [34].

Recombination between repeated elements, including transposable elements, is a well-known source of rearrangement and variation during mitosis and meiosis. Some of these events can lead to dicentric rearrangements and subsequent breakage-fusion-bridge cycles. Furthermore, recombination among intra-chromatid repeated sequences can result in genome rearrangement [6,35]. Depending upon their orientation, sister-chromatid recombination results in the contraction, expansion, or inversion of repeated elements. Recombination between homologs can also generate copy number variations on each diploid homolog, a factor important in human disease [36].

Subtelomeric recombination can also play critical roles in cellular function. One example is the requirement for recombination of genes to subtelomeric regions for antigenic switching and pathogenesis in *Trypanosoma* [37]. Another is a role for subtelomeric recombination in the fidelity of meiotic segregation, as suggested by both cytological and molecular data [38].

Centromeres, the sites of kinetochore attachment, are responsible for proper disjunction of chromosomes in meiosis and mitosis. While the function of the centromere is strongly

maintained, the sequence composition is strikingly variable in different organisms [39]. Centromere repeats in higher eukaryotes are associated with long tracts of repeated satellite DNA that are involved in centromere function and chromatid cohesion [40]. In filamentous fungi, long 30–100 kb AT-rich sequences are present in centromeric regions that are at least in part the consequence of the insertion of TEs followed by RIP [41].

Here I present an investigation of genome rearrangement in *Aspergillus flavus*, based initially on the characterization of large indels between genomes of multiple strains. A large fraction of inserted alleles consists of, or includes, a region of $\geq$ 2 kb AT-rich sequences, defined as regions of $\geq$70% A+T. These non-coding AT elements (ATEs) were studied further in three well-sequenced strains (NRRL 3357, CA14 and SU-16), with each strain having different ATE sizes, chromosomal positions, and degrees of repetition. Based on the comparison of these patterns in different strains, I identified three categories of AT elements. Two of these ATE categories are hot spots for transposable element integration, inactivation, and elimination in either 1) subtelomeric-enriched regions or 2) centromeric domains. ATEs in centromeric regions are highly conserved in position and identity among homologs. I also found a third novel type of subtelomeric and telomeric-adjacent ATEs, lacking detectable TEs, that undergoes frequent homologous recombination during evolution. I discuss data that support the generation of novel centromeric and subtelomeric functions from mutated TE-generated ATEs during *A. flavus* evolution.

## Materials and methods

### Strains

The 9 strains used for my initial analysis were NRRL 3357 (GCA 009017415.1, GCA 014117465.1), CA14 (GCA 014784225.2), AF36 (GCA 012897275.1), AF13 (GCA 014117485.1), K49 (GCA 012896705.1), K54A (GCA 012896555.1), Afla-Guard (GCA 012896875.1), Tox4 (GCA 012896145.1), and A9 (GCA 012895975.1) [42–46]. The data were mined from the most recent sequence version of each strain in the NCBI database. Data from four additional strains, NRRL 2999 (GCA 012897115.1), A1 (GCA 012896995.1), VCG1 (GCA 012896415.1), and VCG4 (GCA 012896275.1) [45], were generated by BLAST homology searches. In addition, SU-16 (GCA 009856665.1) was used for subsequent ATE analyses. The *A. oryzae* strain KBP3 (accession CP031426) and *A. sojae* strain SMF134 (GCA 008274985.1) [47] were also used to study the evolutionary relationships among H class elements. The CA14 assembly was reconstructed by Canu v2.2 [48] to generate the entire telomere-telomere chromosomal sequence. The resulting sequence differed from the previous assembly only by the presence of extensions at telomeres. The Canu assembly was previously successful in the generation of a complete telomere-to-telomere chromosomal sequence of NRRL 3357.

### Identification of genome wide large indels

Chromosomal alignments and images were generated using Lasergene MegAlign Pro 7.2 software (DNASTAR, Inc). The MegAlign program uses a Mauve algorithm [49] to analyze alignment between homologous chromosomes in pairs of strains, excluding centromeric sequences that are only partially sequenced in many strains. Insertions larger than 6 kb were identified and the sequence exported into BLAST 2.0 and filtered using restrictive criteria (97% identity, 50% coverage) to screen for insertions in allelic sequences. Four strains used in these studies did not contain scaffold assembly gaps: NRRL 3357, AF13, CA14 and SU-16. The remaining strains required nucleotide sequence analysis of the putative insertion junctions in each pair analyzed. Putative indels containing scaffold gaps that surround the junctions were excluded from the analysis. Since scaffold gaps are more common in AT-rich regions, this analysis may

underestimate the number of indels. The AT content across each insertion was determined using SnapGene (Dotmatics, Inc).

## Quantitative analysis of large insertions

I used the two complete telomere-to-telomere sequences from strains NRRL 3557 and CA14 to assay the frequency of large insertions ($\geq$ 6kb), walking across each chromosome using the MegAlign procedure. Each insertion was selected and exported into a BLAST 2.0 program, filtering for regions with 90–100% identity. Phylogenetic trees were determined by Fast Minimum Evolution with a maximum sequence difference of 0.2–0.3, as noted. CA14 has a chromosome 3/5 translocation [43]. For simplicity, I report CA14 positions as the corresponding coordinates in the NRRL 3357 genome.

## AT element identification

I used three strains having the most complete sequences, NRRL 3557, CA14, and SU-16, to assay the frequency of AT elements ($\geq$ 2kb), proceeding from left to right arm across each chromosome, using the MegAlign procedure. For AT-rich elements, BLAST 2.0 parameters were filtered, using the following restrictions: a) identity scores $\geq$ 80%, b) significance $< e^{-100}$, c) gap scores of $\leq$ 2%, d) bit scores $\geq$ 1000, and normally no less than 25% of a 100% homology score.

This analysis eliminates weak similarities in favor of a restricted subset of the most significant identities. As a control for potential sequencing errors, I used a second independently sequenced NRRL 3357 [46] to confirm results. No differences were observed between the two sequenced NRRL 3357 genomic sequences.

Importantly, the telomere repeats of SU-16 sequences are present only on chromosomes 4L, 4R, 5L, 5R, and 6L ends. AF13, used for studies of the H class repeats, has intact telomeres at 3L, 4L, 4R, 5L, 6R, 7L, 8L and 8R. Since the left and right orientations of the chromosomes differed in the NCBI database, the arm designations are presented relative to the NRRL 3357 orientation.

## Centromere AT-rich sequence analysis

The AT-rich sequence at each NRRL 3357 centromere was separated into two halves and analyzed by BLAST 2.0 against strains SU-16 and CA14. The centromeres in each of the three strains have been fully sequenced. To estimate the relative stability of centromeric sequences on homologous chromosomes, I determined the highest BLAST score between pairs of homologs, recorded both the observed identity and the associated gap score, and calculated the fraction of centromeric sequence covered by the two regions of homology.

## Censor/Repbase analysis

Analyses of TE-homologous sequences in both ATEs and centromeres were conducted using the Censor program, filtering for fungal species [50]. I restricted the selection of homologies from the Censor read-out using the following parameters: a) mm/Ts (total mismatch mutations/transition mutations) of $<$ 2.2 and b) a bit-score value of $>$ 1000. Both partial and full-length transposable elements fulfilling these criteria were identified and further analyzed. This analysis was extended through BLAST searches for homology to AT-rich derivatives of Gypsy 1, Gypsy 2, and Gypsy 4, first identified by Censor.

### RIP modeling of Gypsy element mutation into ATEs

The sequence divergence and high abundance of transition mutations between target and query sequences confounded the identification of diagnostic dinucleotide pairs in RIP [27]. Similarly, programs such as RIPper [51], used to identify large regions of sequence undergoing RIP, were not informative, due to the rather short size of the regions analyzed and the apparent evolutionary distance from the original RIP activity. I therefore modeled a situation to approximate evolutionary change by assaying the ability of RIP to convert a consensus Gypsy element, as a putative evolutionary progenitor (P), into the observed ATE sequence (A). I considered two expectations for RIP process. First, "P" and "A" differences induced by RIP should consist predominantly of transition mutations, i.e., exhibit a high ratio of transition/total mutations. Thus, the reciprocal of this value, which is a standard Censor output measurement, should be close to 1. Second, the transition mutations noted above should predominantly consist of a cytosine residue in "P" mutating to a thymidine (and the corresponding guanine to adenine mutation on the complementary strand to form "A". The directionality of RIP-induced transition mutations from putative progenitor to ATE Gypsy homologies was estimated by $[(pG > aA) + (pC > aT)]/[(pG > aA) + (pC > aT) + (aC > pA) + aG > pA]$. An unbiased direction of mutation would yield a score of 0.5, while a complete G>A and C>T set of transitions would generate a value of 1.

### Measurement of Gypsy element homology at centromeric and non-centromeric locations

The levels of identity of Gypsy 1, Gypsy 2, and Gypsy 4 elements were determined by BLAST analysis using the Gypsy 1 homologs of NRRL 3357 ATEs 1–6, NRRL 3357 ATE 2–2, and CA14 ATE 3–2; the Gypsy 2 homologs of NRRL3357 ATE 1–2; and the Gypsy 4 homologs of NRRL 3357 ATE 3–1. Both identity and gap levels were recorded, and the overall mean level values and standard deviations were calculated. In the case of centromere Gypsy repeats, I determined the identity among conserved clusters of elements at similar positions on chromosome homologs in each strain. When the Gypsy element is not present within a cluster, I identified the highest scoring homology within the same centromere. Identities between centromeric non-clustered Gypsy copies were lower than the threshold used for the identification of ATEs, and they include percentages as low as 79% and gapped regions as high as 9%.

## Results

### Genome variations measured through indel analysis in *Aspergillus flavus*

The repeated elements within eukaryotes reflect the activities of both recombination and transposition. In *A. flavus*, there is evidence for recombination and genome rearrangement within subtelomeric regions close to the 120 bp of TTAGGGTCAACA telomere repeats [52,53]. However, little is known about the dispersion of transposable and other repeated elements that contribute to genomic stability through the evolution and divergence of this species. I approached this topic through an in silico analysis of unique and repeated element stability in *A. flavus* chromosomes, taking advantage of the many genome sequences available in different strains.

As a first approach to this problem, I identified regions of homology between the chromosomes of 10 sequenced *Aspergillus flavus* strains present at allelic positions in some strains but absent in others. I use the term "large indels" to operationally describe these regions. I conducted a pairwise visual assessment of homologous chromosomes between different strains using the Mauve alignment algorithm of MegAlign Pro 7.2. I sought indels of 6 kb or greater

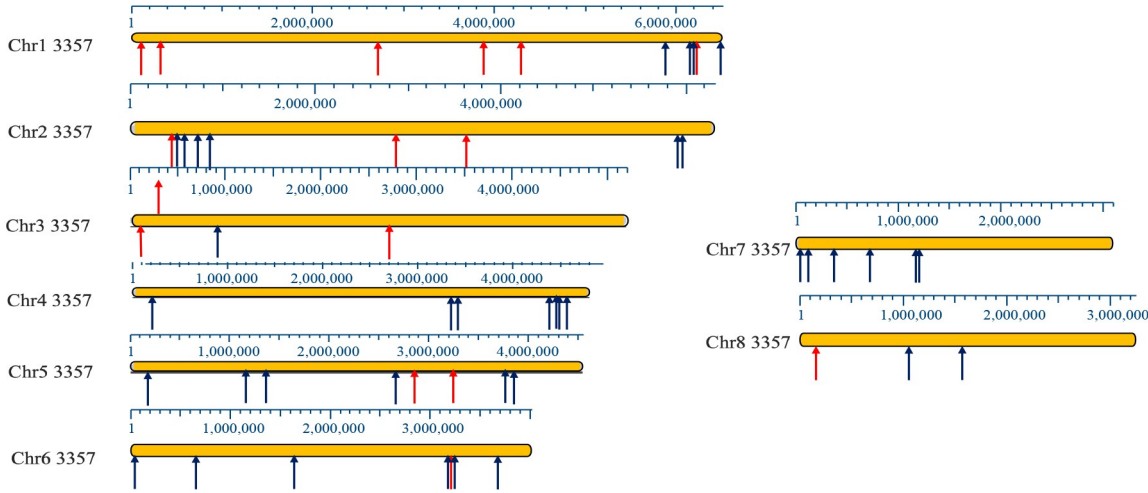

**Fig 1. Distribution map of unique indels in *A. flavus* chromosomes.**

as an initial qualitative screen. I verified and extended the MegAlign results by selecting the region of the putative insertion and exporting the sequence directly into BLAST 2.0. The BLAST search results confirmed the insertions identified by the pairwise analysis. This search also identified the presence or absence of sequences in additional strains not initially analyzed. This analysis excluded centromeric regions that were poorly sequenced in many of the strains. I identified 54 unique insertions, ranging in size from approximately 6.1 to 797 kb at multiple sites at each chromosome [Fig 1 (drawn relative to NRRL 3357 chromosomes); S1 Table].

Non-AT (black) and AT (red) position of deletions or insertions ≥ 6 kb in the 13 strains analyzed, converted to NRRL 3357 coordinates. The data is derived from the S1 Table. Scales in kb are drawn above each chromosome image. Note that the right arm of chromosome 7 is not presented in full since the repeated rDNA genes have been refractory to DNA sequencing. One insertion was previously identified [46].

## AT-rich insertions form a significant fraction of insertions in *A. flavus*

Base composition analysis of the indels revealed the presence of 1500 bp to 31,125 bp regions containing high AT base compositions in 29% of the 54 large insertions scattered throughout the genome (Fig 1, red arrows; S1 Table, red font; S2 Table). The length and base composition of tracts within the insertions are displayed in S3 Table.

I measured quantitatively the frequency of insertions containing either AT elements or non-AT elements in strains NRRL 3357 and CA14. Since the genomes of both strains have a complete sequence of each chromosome (except for chromosome 7R, which contains reiterated rDNA repeats), I performed a telomere-to-telomere analysis of large indels by MegAlign chromosomal walking. I identified large insertions on each chromosome with an equal distribution in both strains. Of the 39 insertions identified, 15 (38%) contained extended regions of AT-rich DNA (S4 and S5 Tables). These AT insertions were enriched within 400 kb of the telomere (8/15; 53%) (Fig 2). Continuous AT-rich sequences ranged in size from 400 bp to 14 kb (S5 Table). All non-AT insertions contained coding regions, as annotated in NRRL 3357 and CA14, that were absent from AT-rich sequences.

Indel quantification in CA14 and NRRL 3357 is summarized in this map, drawn as insertions or deletions relative to NRRL 3357. AT indels are shown as light (deletions, below) or

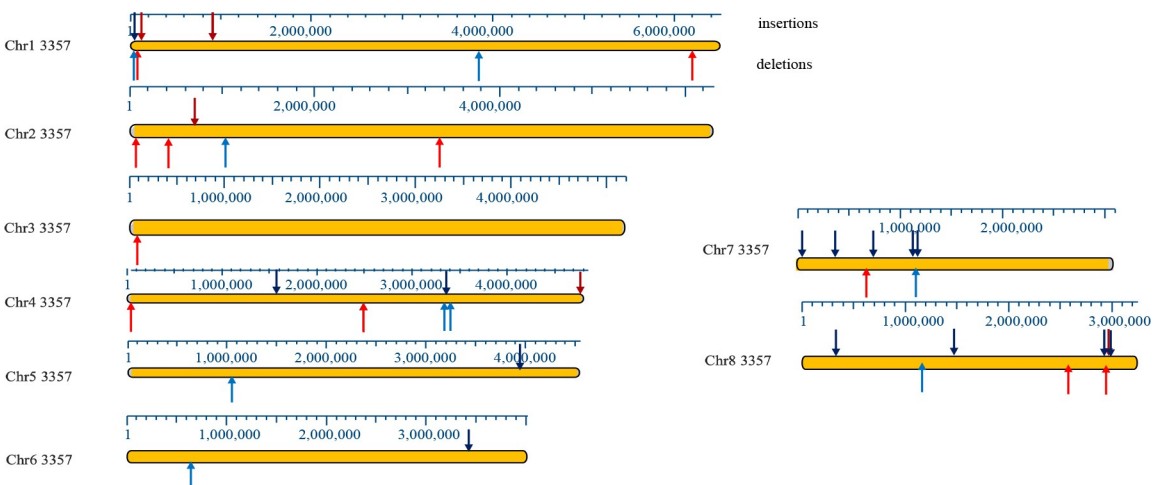

**Fig 2. Telomere-to-telomere analysis of CA14/NRRL 3357 AT indels in *A. flavus* chromosomes.**

dark red (insertions, above) arrows, and non-AT indels are shown as blue (deletions, below) or black (insertions, above) arrows. CEN 3 and CEN5 are transposed due to the translocation between chromosomes 3 and 5 in CA14. In comparisons of NRRL 3357 and CA14, I used the NRRL 3357 nomenclature for simplicity. Scales in kb are drawn above each chromosome image.

## Phylogenetic comparisons reveal dispersion of and selection for AT insertions

To probe the evolutionary characteristics of AT and non-AT insertions, I conducted a phylogenetic comparison of insertions with the tree rooted in either the CA14 or the NRRL3357 strain, depending on which exhibited the inserted sequence [Table 1, S1 and S2 Figs (Figure examples are both rooted in CA14)]. Analysis of two parameters, the insertion dispersion and loss rates, led to two significant findings.

First, non-AT insertions from the genomes of related strains were most often located (in 20/24 cases) at allelic positions in non-AT insertions, when compared to either the NRRL 3357 or CA14. Only 4 out of the 24 non-AT insertions had homology on other chromosomes. In striking contrast, all AT insertions had homologs on non-homologous chromosomes.

Second, I examined the number of strains that lost either the AT or non-AT insertion. The non-AT insertions were absent (at 85% homology) in an average of 5.6 strains, and varied widely from 0 to 12. The AT insertions exhibited an average number of 6.7 strains that had lost the element, ranging from 4–8 strains among the various insertions. These data suggest that while non-allelic dispersion took place primarily in AT insertions, both insertion classes were maintained in the population at similar frequencies. This finding suggests an evolutionary constraint against the loss of the long AT-rich tracts.

## AT elements are clustered in subtelomeric domains

To gain a better understanding of the organization of AT elements (ATEs) independent of indel identification, I analyzed all ATEs that were at least a) 2kb in length and b) 70% A+T, and compared results found in NRRL 3357, CA14, and SU-16 (Table 2A, 2B and 2C). I excluded from this analysis centromeric and pericentromeric sequences (within 30 kb of

**Table 1. Phylogenetic comparisons between NRRL 3357 and CA14 insertions reveal dispersion of AT insertions, and conservation of non-AT insertions.**

| NRRL 3357 AT Insert # | Chr 1 | Chr2 | Chr3 | Chr4 | Chr5 | Chr6 | Chr7 | Chr8 | # Strains Lacking Insert |
|---|---|---|---|---|---|---|---|---|---|
| 1–1 | X | X | | X | X | X | | X | 5/13 |
| 1–14 | X | | | | | X | | | 5/13 |
| 2–1 | X | X | X | | X | X | X | X | 5/13 |
| 2–4 | X | X | | X | X | | X | X | 7/13 |
| 2–5 | X | X | | X | X | | X | X | 7/13 |
| 3–2 | X | | X | X | X | | | X | 5/13 |
| 4–9 | | | | X | | X | X | X | 8/13 |
| 7–6 | X | X | X | X | X | X | X | | |
| 7–6 | | X | | | | | | | 6/13 |
| 8–7 | | X | | | | | | X | 6/13 |
| 8–9 | | | | X | X | | | X | 5/13 |
| Avg. | | | | | | | | | 6.7/13 |
| **NRRL 3357 Non-AT Insert #** | **Chr 1** | **Chr2** | **Chr3** | **Chr4** | **Chr5** | **Chr6** | **Chr7** | **Chr8** | **# Strains Lacking Insert** |
| 1–2 | X | | | | | | | | 5/13 |
| 1–5 | X | | | | | | | | 10/13 |
| 2–2 | | X | | | | | | | 0/13 |
| 4–2 | | X | | X | | | X | X | 4/13 |
| 4–3 | | | | X | | | | | 12/13 |
| 4–5 | | | | X | | | | | 7/13 |
| 5-3-3 | | X | | | X | X | X | | 4/13 |
| 6–3 | | | | | | X | | | 0/13 |
| 7–2 | | | | | | | X | | 4/13 |
| 8–3 | | | | | | | | X | 4/13 |
| Avg | | | | | | | | | 5.6/13 |

(top) Comparisons of homology between each NRRL 3357 AT Inserts (AT Insert #) and the chromosomal location (Chr1 to Chr8) of each homolog in the 12 strains (other than the CA14 strain used for initial analysis) at allelic sites within the same chromosome in genomes including (at a minimum) NRRL 3357 (Yellow Shaded X), at non-allelic sites in genomes including (at a minimum) NRRL 3357 (Blue Shaded X), or at non-allelic sites exclusively in strains other than NRRL 3357 (non-shaded X). In additional, non-allelic homologs in CA14 is shown as a brown shaded X). The right column represents the number of strains that lost the AT insertion at allelic or non-allelic positions in 13 strains analyzed (excluding the strain-either NRRL 3357 or CA14-that was used for the initial identification of indels). The average strain loss is listed in the bottom row. (bottom) Comparisons of homology between each NRRL 3357 non-AT Insertion (Non-AT Insert #) and each chromosomal homolog (Chr1 to Chr8) [other than the strain used for initial analysis (CA14)) at allelic sites in genomes including (at a minimum)NRRL 3357 (Yellow Shaded X) or at non-allelic sites exclusively in genomes other than NRRL 3357 (non-shaded X). The right column represents the number of strains that lost the non-AT insertion in the 13 strains analyzed. The average strain loss is listed in the bottom row.

centromeric A+T-rich sequences). I found 31 ATEs in NRRL 3357, 27 ATEs in CA14, and 24 ATEs in SU-16. The majority of these were present within the 100 kb region adjacent to the telomere (termed subtelomeres here, as indicated by the yellow shaded entries in the Table) in the two strains that were fully sequenced from telomere to telomere (68% and 59% in NRRL 3357 and CA14, respectively). In NRRL 3357, 9 of the 31 ATEs were directly adjacent to telomeric tracts. AT elements were also enriched adjacent to the telomeres in CA14 (6 of 27 ATEs). Based on a comparison of both the numbers and coverage of ATEs within and outside of subtelomeric regions (excluding centromeric regions), I estimate an ATE subtelomeric enrichment of 26-fold in CA14 and 56-fold in NRRL 3357. Furthermore, although

**Table 2. A. Approximate positions, lengths, AT contents, and classes of NRRL 3357 ATEs.** B. Approximate positions, lengths, AT contents, and classes of CA14 ATEs. C. Approximate positions, lengths, AT contents, and classes of SU-16 ATEs.

| CHR-ATE# | AT ELEMENT Endpoint 1 | AT ELEMENT Endpoint 2 | LENGTH (BP) | TA | %AT | Repeat Class |
|---|---|---|---|---|---|---|
| 1–1 | 9504 | 14645 | 5062 | | 92 | A |
| -2 | 30420# | 41021 | 10602 | | 90 | B |
| -3 | 68529 | 70534 | 2006 | | 74 | F |
| -4 | 73018 | 78079 | 5062 | | 78 | C |
| -5 | 1051356 | 1053170 | 1815 | | 81 | C |
| -6 | 6193134# | 6208748 | 15615 | | 86 | D, E |
| -7 | 6506754 | 6509668 | 2915 | X | 85 | H |
| 2–1 | 329733# | 340907 | 11175 | | 90 | D |
| -2 | 404403# | 414049 | 9647 | | 89 | C. D |
| -3 | 3669069# | 3679001 | 9933 | | 95 | D |
| 3–1 | 75597# | 84574 | 8978 | | 89 | E |
| -2 | 2687253# | 2708073 | 20821 | | 81 | D |
| -3 | 5166534 | 5173314 | 6781 | | 88 | A, C |
| -4 | 5190815 | 5196380 | 5566 | X | 80 | G, H |
| 4–1 | 335# | 17239 | 16905 | X | 89 | H, I |
| -2 | 4796088 | 4798609 | 2522 | | 85 | C |
| -3 | 4806466 | 4809957 | 3492 | X | 84 | F |
| 5–1 | 144 | 6637 | 6494 | X | 92 | H |
| -2 | 4506908 | 4508817 | 1910 | | 78 | C |
| -3 | 4538235 | 4549409 | 11175 | X | 90 | D |
| 6–1 | 239 | 7306 | 7068 | X | 85 | H |
| -2 | 10268 | 22015 | 11748 | | 82 | B, C |
| 7–1 | 239 | 4250 | 4012 | X | 83 | I |
| -2 | 8358 | 10935 | 2578 | | 81 | |
| -3 | 687631# | 693647 | 6017 | | 93 | D |
| 8–1 | 335 | 8739 | 8405 | X | 86 | H |
| -2 | 23257 | 27841 | 4585 | | 72 | J |
| -3 | 3222321 | 3243523 | 21203 | | 90 | E |
| -5 | 2576735# | 2584367 | 7633 | | 92 | L |
| -6 | 2913535# | 2922027 | 8531 | | 86 | E |
| -4 | 3246389 | 3251641 | 5253 | X | 91 | |

| CHR-ATE# | AT ELEMENT Endpoint 1 | AT ELEMENT Endpoint 2 | BP | %AT | TA | Repeat Class |
|---|---|---|---|---|---|---|
| 1–1 | 24976 | 27841 | 2866 | 76 | | C |
| -2 | 686673 | 690588 | 3916 | 91 | | C |
| -3 | 1611755# | 1621496 | 9742 | 92 | | C, E |
| -4 | 2157327 | 2161528 | 4202 | 89 | | C |
| -5 | 5166492 | 5171362 | 4871 | 82 | | K |
| -6 | 5476699 | 5478704 | 2006 | 77 | | C |
| -7 | 6462775# | 6473185 | 10124 | 86 | | E |
| -8 | 6475860 | 6477769 | 1910 | 75 | | F |
| -9 | 6505182 | 6509001 | 3820 | 87 | | |
| -10 | 6525048 | 6535171 | 10123 | 92 | | C |
| -11 | 6546824 | 6557092 | 10269 | 92 | | H |
| 2–1 | 749106# | 763241 | 14135 | 92 | | D, L |
| 3–1 | 105 | 6027 | 5923 | 78 | x | I |

*(Continued)*

| CHR-ATE# | AT ELEMENT Endpoint 1 | AT ELEMENT Endpoint 2 | BP | % AT | | Repeat Class |
|---|---|---|---|---|---|---|
| -2 | 6180384 | 6185254 | 4871 | 91 | | D |
| -3 | 6262238 | 6268206 | 5971 | 90 | x | G |
| 4–1 | (-2186) | 1204 | 3390 | 84 | x | |
| -2 | 4474953 | 4478868 | 3916 | 90 | | K |
| -3 | 4702628# | 4708837 | 6210 | 83 | | C |
| -4 | 4710080# | 4712085 | 2006 | 74 | | |
| -5 | 4714283# | 4717529 | 3247 | 76 | | |
| 5–1 | 3430647 | 3437430 | 6784 | 88 | | A |
| -2 | 3454909 | 3458499 | 3592 | 83 | x | C |
| 6–1 | 426278 | 428855 | 2578 | 73 | | |
| -2 | 4065546 | 4076437 | 10878 | 91 | x | B |
| 7–1 | 8 | 5690 | 5682 | 85 | x | H |
| 8–1 | 177413# | 183047 | 5635 | 77 | | |
| -2 | 3171697 | 3176280 | 4584 | 72 | | J |

| CHR-ATE# | AT ELEMENT Endpoint 1 | AT ELEMENT Endpoint 2 | BP | % AT | Repeat Class |
|---|---|---|---|---|---|
| 1–1 | 18195 | 29273 | 11079 | 90 | D, F |
| -2 | 41400 | 51909 | 10510 | 88 | C, E |
| -3 | 168387 | 176931 | 8545 | 89 | E |
| -4 | 291817 | 299182 | 7366 | 88 | E |
| -5 | 710086 | 719220 | 9135 | 82 | C, D |
| -6 | 4436476 | 4439149 | 2674 | 83 | C |
| -7 | 4525067 | 4527071 | 2005 | 83 | C |
| -8 | 6535277 | 6538532 | 3256 | 74 | C |
| 2–1 | 2725706 | 2732199 | 6494 | 82 | D |
| 3–1 | 4606353 | 4611747 | 5395 | 70 | |
| 4–1 | 4085476 | 4087863 | 2388 | 75 | |
| -2 | 4123029 | 4134107 | 11079 | 83 | B, K |
| 5–1 | (1) | 3008 | 3008 | 91 | |
| -2 | 1460769 | 1467263 | 6495 | 78 | D |
| -3 | 2336734 | 2340554 | 3821 | 91 | C |
| -4 | 4323756 | 4327671 | 3916 | 84 | K |
| 6–1 | 677892 | 679896 | 2005 | 86 | |
| -2 | 3248415 | 3255864 | 7450 | 73 | D |
| -3 | 3454894 | 3465112 | 10219 | 83 | E |
| -4 | 3758602 | 3777512 | 18910 | 87 | D, E |
| -5 | 3821152 | 3826595 | 5444 | 82 | D |
| -6 | 392973 | 3931736 | 2006 | 78 | C |
| -7 | 4197076 | 4199558 | 2483 | 83 | E |
| 8–1 | 1609461 | 1617101 | 7641 | 83 | L |
| -2 | 3129415 | 3132470 | 3056 | 78 | J |

Yellow shading indicates an ATE within the subtelomeric 100 kb of the chromosome. TA refers to subtelomeric ATEs that are directly adjacent to telomeric repeats. Repeat classes of ATEs described in text are listed in the column on the right. Abbreviations: Chr: Chromosome, AT element 1, 2: Approximate NRRL 3357 chromosomal coordinates of ATEs of left and right junctions, respectively, # refers to insertions previously identified by indel analysis between genomes of NRRL 3357 and CA14.

Yellow shading indicates an ATE within the subtelomeric 100 kb of the chromosome. TA refers to subtelomeric ATEs that are directly adjacent to telomeric repeats. Repeat classes are denoted in the right-most column. Chr: Chromosome, AT element 1, 2: Approximate CA14 chromosomal coordinates of ATEs of left and right junctions; # refers to insertions previously identified by indel analysis between genomes of NRRL 3357 and CA14.

Yellow shading indicates an ATE within the subtelomeric 100 kb of the chromosome. The classes of ATEs described in text are listed in the right-most column. Abbreviations: Chr: Chromosome, AT element 1, 2: Approximate SU-16 chromosomal coordinates of ATEs of left and right junctions, respectively.

subtelomeres are only partially represented in SU-16, 25% of AT elements were nonetheless found in this domain. As expected, most of the insertions identified in the indel analysis between CA14 and NRRL 3357 (indicated by the hashtag in Table 2A and 2B) were also found in these more detailed analyses.

Conversely, most subtelomeric regions exhibited clustering of AT elements. Within the NRRL 3357 100 kb subtelomere, areas of at least 70% A+T and 2 kb in size were present at 12 of the 15 assayable ends but absent from both subtelomeres on chromosome 2 and the right arm of chromosome 6. In the revised CA14 sequence (see Materials and Methods), I found that most (9/15) subtelomeric regions contained ATEs. However, the ATEs in CA14 bore no discernable relationship to the ATEs structure in NRRL 3357.

## Identification of multiple classes of homologous AT elements

I surveyed the homologies found between the ATEs within and among the three strains (**Table 2**). The homologies fell into twelve classes (A-L). Reanalysis of AT-element size in NRRL 3357 led to revised size ranges from 1.9 kb-21.2 kb. 29 of the 31 elements fall into several classes representing shared homology (Table 2A; Classes A-J, Class L), as determined by BLAST analysis, with two unique elements. AT elements in CA14 range in size from 2.0 kb-14.1 kb. 21 of the 26 ATEs fall into classes that share homology (Table 2B; Classes B-L) with five unique elements. AT-elements in SU-16 range in size from 2.0 kb-18.9 kb. Eighteen of the 25 ATEs fall into classes that share homology (Table 2C; Classes B-F, J-L), with four unique classes. These data revealed that ATE lengths and chromosomal positions varied significantly among the different genomes.

Each class shows a different degree of association with subtelomeres. Members of one such class (class H) were present adjacent to (or in one case within 3 kb of) telomeric-repeats, while others (Classes A, F, G, I and J) were present within the subtelomere but not always adjacent to the telomere. Members of a third group were enriched, but not present exclusively, at subtelomeres (Classes B-E). These latter classes were present at subtelomeres in 27/51 (52.9%) cases (compared to the predicted value of 4.44%, if distributed randomly in the genome). The full dataset of the homologies among each member of the A-L class is documented in S3 Fig.

## ATEs contain homology to multiple classes of transposable elements

The homologies among ATEs could represent known classes of repeated elements. To study this further, I used the Censor program that examines potential repeated elements within the Repbase database [54]. This database characterizes candidates based on similarity, BLAST score, and the probability that mutations result from gradual evolution, by measuring the prevalence of transition versus transversion mutations. The most significant homologies found in this study represent a diverse series of transposable elements that were previously characterized as consensus sequences in the *A. flavus* ecotype, *Aspergillus oryzae*. I identified both Type 1 and Type 2 transposable element homologs (Tables 2A–2C and 3, S3 Fig). First, I identified an inactivated Type 2 Mariner 1 consensus sequence at multiple ATEs in all three strains. Indeed, the Mariner 1 class accounts for the repeated ATE classes C (S3C Fig). Two other members of the Mariner class (Mariner 5 and Mariner 2N-1) were found less frequently.

The major Type 1 LTR-retrotransposon class consists of Gypsy elements and includes both the retrotransposon coding regions and solo-LTR elements. Solo-LTRs are likely formed through homologous recombination [55], but are still indicative of previous retrotransposition. AT-rich mutated forms of the *A. oryzae* Gypsy elements (designated AO) were identified in *A. flavus* and subsequently used to screen for additional mutated forms of each Gypsy species by BLAST (designated AF). Using these methods, I identified homology to Gypsy 1/Gypsy

**Table 3. Summary of composition of ATE classes in non-centromeric DNA.**

| Class | Origin |
|---|---|
| A | STE |
| B | Gypsy 2 |
| C | Mariner 1 |
| D | Gypsy 1 |
| E | Gypsy 4 |
| F | STE |
| G | STE |
| H | STE |
| I | STE |
| J | STE |
| K | UK |
| L | UK |

Abbreviations: Origin, source of ATE homologies; STE, subtelomeric ATE elements, UK, unknown.

1 LTR in class D ATEs, to Gypsy 2/LTR1 in class B ATEs, and to Gypsy 4-AFLAV/Gypsy 4 LTR in class E ATEs, (Table 3, S3B, S3D and S3E Fig). In all cases, the presence of the relevant TE fully explains the class B-E homologies. BLAST analyses also revealed that LTR1 is an AT-rich mutated form of Gypsy 2 LTR and that AFLAV [10] is the Gypsy 4–1 homolog of *A. flavus* (data not shown). Another putative Type 1 retrotransposon class, LTR2, was defined solely by its LTR repeat. The composition of each class is summarized in Table 3. Each strain had comparable levels of Mariner and Gypsy homologs as measured by abundance and density (# copies/Mb ATE) (Table 4A and 4B). The level of conservation between Gypsy elements in the non-centromeric ATEs is shown in Table 4C.

The only non-LTR retrotransposon homologs were highly mutated TAD1 [56] species. Hence, retrotransposition events are evolutionarily frequent, with a different array predominating in each of the three strains.

## ATE transposable elements show evidence of evolutionary RIP-like activity

The fungal repeat-induced point mutation (RIP) system inactivates functional repeated sequences, including TEs. This process involves the coupled activity of cytosine methylation and C>T transition mutations. RIP activity in some fungi can be influenced by the local context of the mutated cytosine residues and ultimately gives rise to AT-rich sequences. Hence, a region containing multiple TEs can mutate into an AT-rich region. I sought evolutionary footprints for this process to determine the nature of RIP activity in ATEs. RIPper and RIPCAL [51,57], two algorithms designed for measuring likely RIP frequencies over genomic distances, do not provide interpretable information for the relatively short and highly mutated sequences under analysis here. Hence, I modeled RIP-like activity by considering the preferential accumulation of C>T and G>A transition mutations necessary to convert a TE consensus sequence into the ATE TE homologs. For this analysis, I compared Gypsy 1, Gypsy 2, and Gypsy 4 TEs, since the reference database of these elements had intact coding regions, unlike many mutated consensus sequences present in Repbase. Two criteria were measured. First, the ratio of transitions/total mutations was measured by Censor analysis. The average Gypsy 1, Gypsy 4 and Gypsy 2 total mutations/transition mutations indicated that transversions consisted of less than 0.01% of all mutations (S6 Table). Gypsy LTRs gave similar results. Hence, all three of these classes showed clear evidence of RIP.

**Table 4. Characterization of Gypsy and Mariner TEs in *Aspergillus flavus* strains.** A. Distribution of non-centromeric TE homologs. B. Density of TEs in centromeric and non-centromeric ATEs. C. Homology among Gypsy elements in ATEs.

| Element | NRRL 3357 | CA14 | SU-16 |
|---|---|---|---|
| **Gypsy 1** | 10 | 3 | 8 |
| **Gypsy 2** | 3 | 3 | 1 |
| **Gypsy 4** | 5 | 2 | 6 |
| **Gypsy Solo LTRs** | 3(1); 9(2); 7(4) | 4(2); 7(4) | 1(1); 2(2) |
| **Mariner 1** | 10 | 12 | 10 |
| **Mariner 2N1** | 1 | 0 | 2 |
| **Mariner 5** | 1 | 1 | 0 |
| **LTR 2** | 4 | 4 | 1 |
| **TAD 1** | 1 | 0 | 2 |

| Strain | NRRL 3357 | NRRL 3357 | CA14 | CA14 | SU-16 | SU-16 |
|---|---|---|---|---|---|---|
| **Element** | NC ATE normalized | Cen ATEs | NC ATE normalized | Cen ATE | NC ATE Normalized | Cen ATE |
| **Gypsy 1** | 35 | 0 | 17.4 | 1 | 40.88 | 0 |
| **Gypsy 2** | 10.5 | 6 | 17.4 | 7 | 5.11 | 4 |
| **Gypsy 4** | 17.5 | 11 | 11.6 | 8 | 30.66 | 10 |
| **Mariner 1** | 35 | 14 | 69.6 | 10 | 51.1 | 12 |

| Element | Sample Size (n) | Average Homology (%) | SD | Average Gap (%) | SD |
|---|---|---|---|---|---|
| **Gypsy 1** | 18 | 88.7 | 4.8 | 0.96 | 1.5 |
| **Gypsy 2** | 4 | 85.7 | 2.5 | 1.7 | 1.7 |
| **Gypsy 4** | 14 | 89.8 | 2.5 | 0.8 | 1.5 |

The number of identified homologs to TEs elements (from Table 2A–2C). Gypsy Solo LTRs are specified for Gypsy 1 (1), Gypsy 2(2) or Gypsy 4(4).

The number of Gypsy and Mariner 1 homologs in non-centromeric ATEs (NC-ATEs) per genome were corrected to the expected size ATEs, normalized to the combined sizes of centromeric DNAs, and compared to homologs observed in centromeric ATE (Cen ATEs).

Average homology and gap percentages are presented with standard deviations for each element in the three strains, NRRL 3357, CA14 and SU-16. n, the number of samples when homology is measured relative to the Gypsy 1 element in NRRL 3357 ATEs 1–6, NRRL 3357 ATE 2–2, or CA14 ATE 3–2, relative to the Gypsy 2 element in NRRL 3357 ATE 3–2, or relative to the Gypsy 4 element in NRRL 3357 ATE 3–1.

Second, using this model, I measured the fractions of all transitions that are the consequence of a unidirectional mutation of the consensus Gypsy elements to form the corresponding ATE Gypsy homologs (see Materials and Methods). A ratio close to 1 indicates a pattern consistent with the G to A and C to T mutational consequence of RIP. Indeed, values ranged from 0.8 to 1.0, arguing in favor of RIP activity over evolution.

## Chromosome-specific centromeric ATEs are highly conserved

Filamentous fungi contain conserved features of the eukaryotic kinetochore. However, the centromeric sequences are embedded within large > 30 kb A+T-rich regions (~100 kb in *A. flavus*) [41]. Given the variation in the position and identity of non-centromeric ATEs in evolution, I asked whether centromeric AT-rich regions were similarly unstable (Table 5). I therefore tested the extent of contiguous homology between centromeres in NRRL 3357 and either CA14 or SU-16. I found the two longest stretches of contiguous homology in SU-16 and CA14 chromosome 1 centromeres, which are 95% identical to NRRL 3357 centromeres over 56% of the length. Centromeres of all chromosomes exhibited similar characteristics. In SU-16, the coverage ranged from 56% on chromosome 1 to 100% on chromosome 4, with identities to

**Table 5. Centromere sequences of homologs have strongly conserved regions.**

| STRAIN CEN | NRRL3357 (bp assayed) | SU16 Longest Region of Homology (bp/%) | %Identity/~%Gaps | CA14 Longest Region of Homology(bp/%) | %Identity/~% Gaps |
|---|---|---|---|---|---|
| 1 | 124349 | 70358/56.6 | 95.2/2 | 69522/55.9 | 96/1 |
| 2 | 108565 | 93858/86.4 | 93.4/2.5 | 108851/100.2 | 99/0 |
| 3 | 109188 | 92730/84.9 | 97.7/1 | 705626/64.6 | 95.2/1.5 |
| 4 | 101074 | 102524/101.4 | 94.1/2.5 | 47291/46.8 | 92.5/3.5 |
| 5 | 117326 | 103407/88.1 | 96.4/1.5 | 94696/80.7 | 97.9/0 |
| 6 | 122829 | 99959/81.4 | 92.7/2.5 | 122744/99.9 | 99/0 |
| 7 | 97528 | 90869/93.1 | 98.5/0 | 75338/77.2 | 99/0 |
| 8 | 108717 | 79595/73.2 | 91.5/4.0 | 81048/74.5 | 99/0 |

Homology between NRRL 3357 centromeres 1 through 8 and either CA14 or SU-16 centromeres was determined by BLAST analysis of left-arm proximal and left arm distal halves of each centromere, and the data was summed. The longest regions of homology are presented in bp and as a percentage of the overall NRRL 3357 sequence analyzed in each strain, together with identity and gap scores. Gap scores are rounded to the nearest 0.5 value.

NRRL 3357 ranging from 88% to 99%. For CA14, the coverage ranged from 47% in chromosome 4 to 100% in chromosomes 2 and 6, with identities to NRRL 3357 varying between 92% and 99%. Given the inability to check the centromeric sequences against a sequenced repeat, I cannot rule out the presence of sequencing errors that may lower the degree of homology. No significant similarity was observed between the centromeric ATEs of non-homologs (data not shown).

## Conservation of Gypsy elements on homologous chromosomes

Given the conservation of the centromeres in homologous chromosomes, I mapped the relative positions of Gypsy elements in the homologs. This analysis revealed a conserved clustering in the three major strains analyzed, although insertions and deletions of elements are still present (Figs 3 and S4). BLAST and sequence analysis revealed that Gypsy 2 and Gypsy 4 elements

**Fig 3. Organization of Gypsy element homologs at chromosome 4 centromeres in NRRL 3357, SU-16 and CA14.**

in these clusters were highly conserved in different strains on chromosomes 1, 4, 6 and 8 (Table 6A and 6B), S5A and S5B Fig). In contrast, centromeric Gypsy 2 and Gypsy 4 elements displayed less similarity at non-clustered unique sites (79–90%) and were often highly gapped (up to 9%), suggesting a pressure for centromeric homology early in evolution of the strains. Indeed, even Gypsy elements in non-centromeric ATEs had a generally higher average level of homology than between non-clustered centromeric Gypsy elements [compare Table 4C with Table 6A and 6B]. Gypsy 1 elements, on the other hand, were only present at one centromeric site, with two others found in pericentromeric regions (data not shown).

The positions of the mutated Gypsy 2 and Gypsy 4 homologs within the centromeric domains (between the coordinates listed) were plotted and aligned among the three strains. The red, green, blue, and brown colors refer to regions of high homology as shown in Table 6. The white color refers to regions of lower homology (corresponding to the black color in Table 6A). The GC content of these regions is presented above each chromosomal plot. The coding regions flanking the centromere are depicted by the purple arrows. The sequence range displayed for each strain was edited to align centromere sequences.

Curiously, I found quantitative differences among some transposition classes in centromeric and non-centromeric within the ATE sequences (Table 4B). Specifically, there is a marked increase in the density of non-centromeric Gypsy 1 and Mariner 1 elements relative to centromeres, suggesting that non-centromeric ATEs either represent relative hot spots for these TEs, or TEs are more stable in a non-centromeric context.

## Highly conserved subtelomeric repeats form unique classes of ATEs

Homologs to a consensus sequence of highly mutated Mariner 1 and Gypsy elements account for four of the repeat classes (Classes B-E). However, the homology between other termini is qualitatively different from these classes, apart from the failure to find an identifiable TE homolog. First, one class (Class H) lies directly adjacent to telomeric repeats (except in CA14 ATE 1–11, which lies 3 kb from the telomere). Five additional repeats (Classes A, F, G, I and J) are in the subtelomeric region, but not always adjacent to the telomere. None of these repeats are present within centromeric DNA.

Second, in each of these classes (Classes A, F-J), the polarity with respect to the termini is constant (Figs 4, S3A and S3F–S3J), resulting in inverted repeats on the two ends of a given chromosome. This differs from the typical random orientation of TE insertions.

Examples of the position and polarity of H class elements at the subtelomeres, including NRRL 3357 ATE 5–1 (left arm) and ATE 8–4 (right arm), AF13 Chr7 STLA (subtelomeric left arm telomere-adjacent), and CA14 ATE 1–11 (right arm), as defined by homology to NRRL 3357 ATE 4–1 (blue filled line). The nucleotide position of homology in ATE 4–1 homology is given in parentheses. The polarity in each case (shown by the arrowhead) is identical with respect to the telomere (drawn for simplicity as moving away from the telomere) regardless of chromosomal arm.

Third, Class H elements can be explained fully by the recombination of a putative progenitor with homology to NRRL 3357 ATE 4–1. This repeat is itself an imperfect direct repeat of an element most homologous to NRRL 3357 ATE 6–1. We observed multiple patterns of rearrangement of NRRL 3357 ATE 4–1 among the different strains, with some strains maintaining the ATE 4–1 structure (e.g., NRRL 3357 and AF13) and others deleted for part of the sequence (e.g., CA14, SU-16, and the *A. oryzae* strain KBP3) (Fig 5). The finding of ATE 4–1 in two normally separated strains suggests the conserved presence of ATE 4–1 in the evolutionary progenitor that subsequently underwent different patterns of loss and rearrangement during the formation of the various strains. Further supporting this view is the common presence of

**Table 6. Homology of Gypsy 4 elements at clustered and non-clustered sites in chromosomes 1, 4, 6, and 8.** A. B. Homology of Gypsy 2 elements at clustered and non-clustered sites in chromosomes 4, 6, and 8.

| Chr | Strain | EP 1 | EP 2 | NRRL 3357 | | CA14 | | SU-16 | |
|---|---|---|---|---|---|---|---|---|---|
| | | | | % homology | % gaps | % homology | % gaps | % homology | % gaps |
| 1 | NRRL 3357 | 4612525 | 4619143 | 100 | 0 | 90 | 0 | 87 | 2 |
| | """ | 4537928 | 4544535 | 100 | 0 | 99 | 0 | 99 | 0 |
| | """ | | | | | | | | |
| 4 | NRRL 3357 | 2791537 | 2797647 | 100 | 0 | 80 | 9 | 99 | 0 |
| | """ | 2788942 | 2790878 | 100 | 0 | 97 | 1 | 99 | 0 |
| | """ | 2798042 | 2799269 | 100 | 0 | NS | | 98 | 2 |
| | """ | 2799236 | 2802975 | 100 | 0 | 95 | 3 | 98 | 1 |
| | | | | | | | | | |
| | CA14 | 2765037 | 2765872 | 78 | 7 | 100 | 0 | 79 | 8 |
| | | | | | | | | | |
| | SU16 | 2686687 | 2693294 | 83 | 4 | 84 | 2 | 100 | 0 |
| | | | | | | | | | |
| 6 | NRRL 3357 | 2162182 | 2165127 | 100 | 0 | 99 | 0 | 99 | 0 |
| | """ | 2158636 | 2161862 | 100 | 0 | 98 | 0 | 99 | 0 |
| | | | | | | | | | |
| 8 | NRRL 3357 | 1825191 | 1829507 | 100 | 0 | 99 | 0 | 86 | 6 |
| | """ | 1829775 | 1831639 | 100 | 0 | 99 | 0 | 97 | 2 |
| | """ | 1888227 | 1891520 | 100 | 0 | 99 | 0 | 94 | 4 |
| | | | | | | | | | |

| Chr | Strain | EP 1 | EP 2 | NRRL 3357 | | CA14 | | SU-16 | |
|---|---|---|---|---|---|---|---|---|---|
| | | | | % Homology | % gaps | % homology | % gaps | % homology | % gaps |
| 4 | NRRL 3357 | 2803656 | 2809455 | 100 | 0 | 98 | 0 | 99 | 0 |
| | """ | 2731528 | 2736930 | 100 | 0 | 99 | 0 | 97 | 1 |
| 6 | NRRL 3357 | 2154077 | 2157751 | 100 | 0 | 99 | 0 | 99 | 0 |
| | """ | 2201360 | 2203545 | 100 | 0 | 98 | 1 | 99 | 0 |
| | """ | 2165791 | 2165791 | 100 | 0 | 99 | 0 | 99 | 0 |
| 8 | NRRL 3357 | 1838843 | 1843069 | 100 | 0 | 99 | 0 | 81 | 7 |
| | | | | | | | | | |

Gypsy 4 elements within each centromere were tallied and the approximate coordinate endpoint (EP) positions noted. Each of the fragments was measured for homology to repeats at clustered sites (bold) in the three strains, NRRL 3357, CA14 and SU16. The second column refers to the strain containing the Gypsy element used for comparison in the analysis. When a clustered site is not present in one of the strains, the highest scoring fragment is listed (plain text). The Gypsy elements present in clusters in two to three of the strains are shown in red, green, blue, or brown hues, corresponding to the colors in Fig 4 and S4 Fig. The Gypsy elements, not present in a cluster in any of the strains (black) are presented in a white hue in the corresponding figures; NS, no significant homology detected.

Gypsy 2 elements within each centromere were tallied and the approximate coordinate endpoints (EP) positions noted. Each of the fragments was measured for homology to repeats at clustered sites (bold) in the three strains, NRRL 3357, CA14 and SU16. The second column refers to the strain containing the Gypsy element used for comparison in the analysis. When a clustered site is not present in one of the strains, the highest scoring fragment is listed (plain text). The Gypsy element present in clusters in two to three of the strains are shown in red, green, or blue hues, corresponding to the colors in Figs 3 and S4; NS, no significant homology detected.

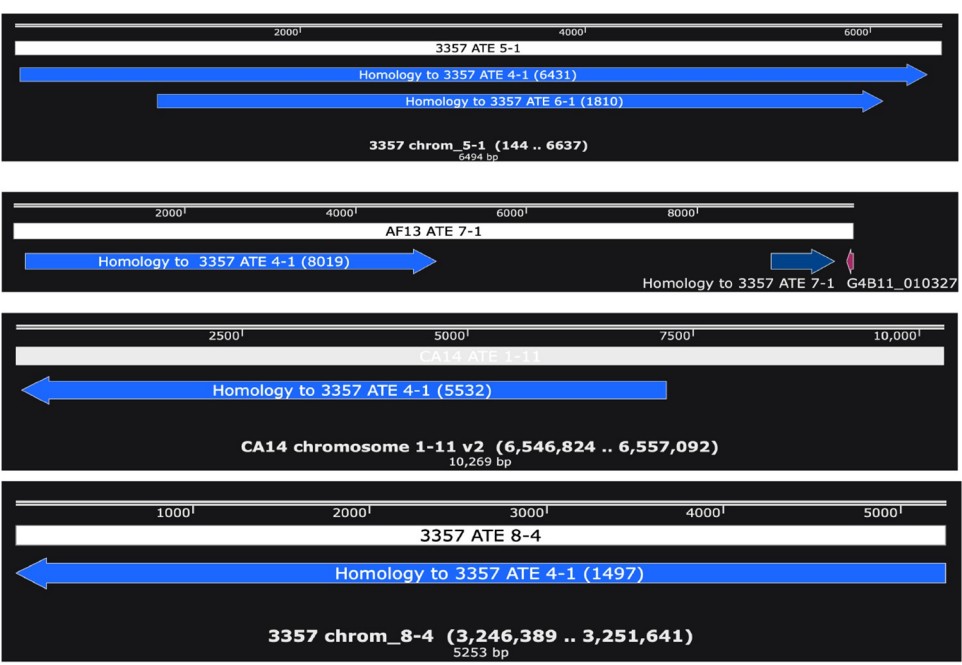

**Fig 4. Defined polarity of H class elements.**

downstream TEs and homologies, both in the subtelomeric adjacent sequences of chromosome 4L in AF13 and NRRL 3357 and at other flanking sequences of H class species (S3H Fig).

The regions of homology among different H class sequences relative to the NRRL 3357 4–1 ATE. The different colors correspond to different strains: purple, AF13; green, *A. oryzae* strain KBP3; blue, NRRL 3357; red, CA14. The AF13 Chromosome 4L telomere is 99% identical to NRRL 3357 ATE 4–1.

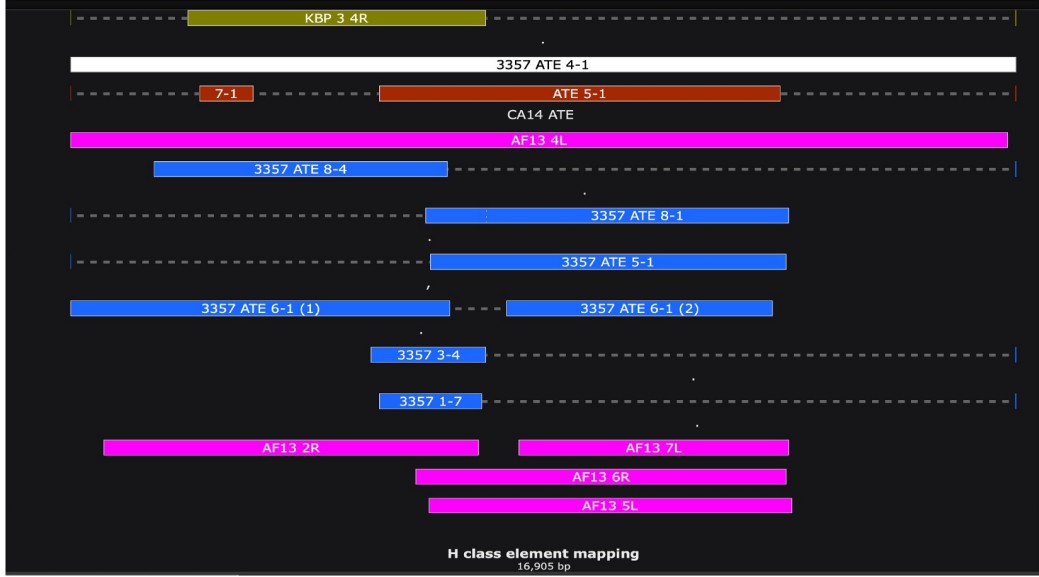

**Fig 5. Relative positions of H class elements.**

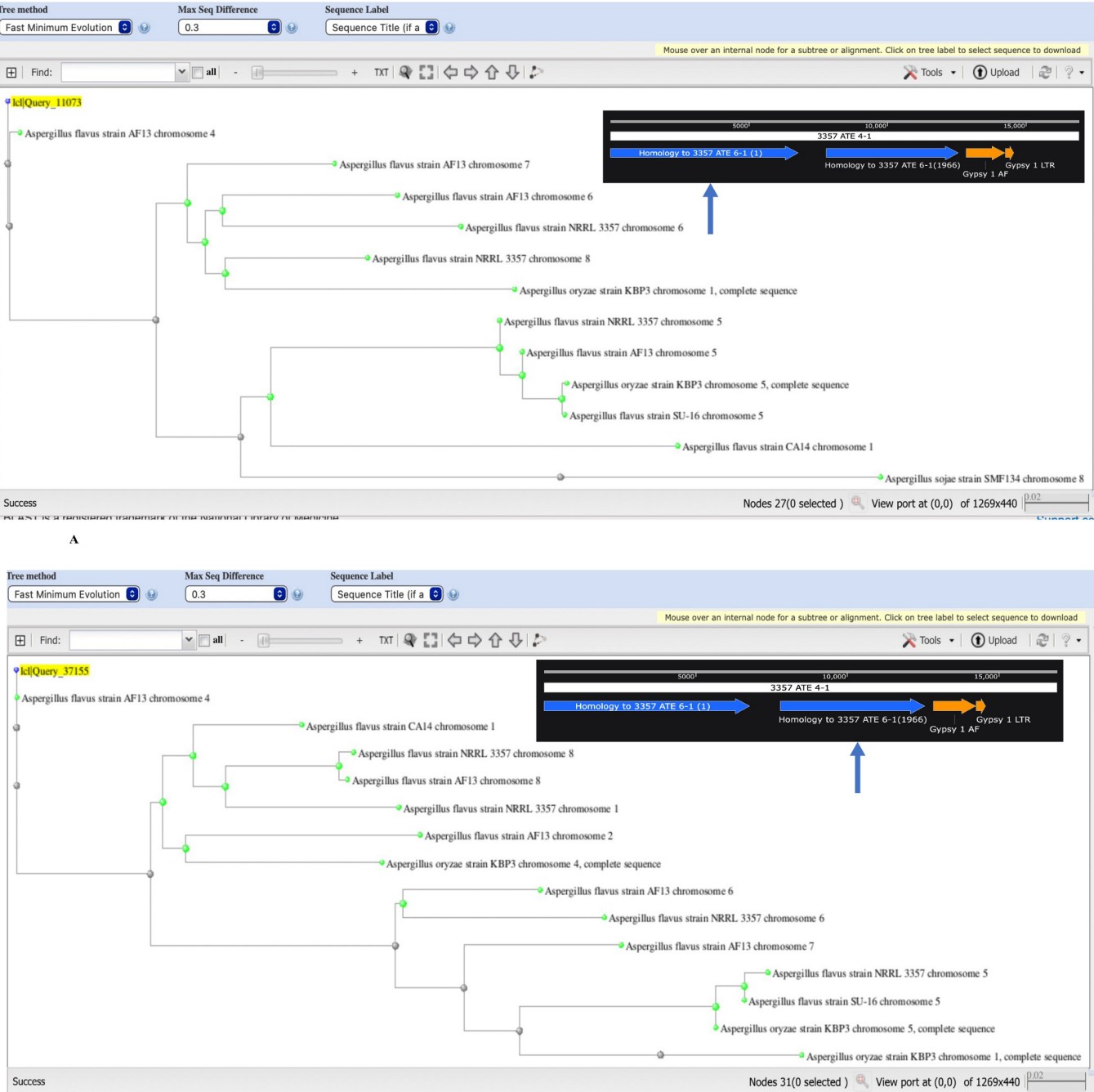

**Fig 6. Evolutionary relationships between the left and right class H repeats.**

To test this evolutionary relationship further, I used the BLAST Tree Widget Profile with the Fast Minimum Evolution algorithm to compare the two adjacent, but divergent, sub-repeats present in the NRRL 3357 4–1 fragment, identified by homology to NRRL ATE 6–1 (Fig 6). Comparison of the two sub-repeats to the elements scattered at different subtelomeric loci revealed that each half is evolutionarily related to different H class homologs. What is also evident is that this homology is maintained (albeit at lower levels) even in the *A. oryzae* and *A.*

*sojae* strains of the *Flavi* clade. This analysis suggests that different portions of the NRRL 3357 ATE 4-1-like imperfect dimer precursor diverged during the formation of the different strains to give rise to the current array of repeats. These results argue for an evolutionarily robust recombinational activity at these telomere-adjacent loci.

BLAST Tree relationships of different strains with the left (A) and right (B) repeats of NRRL 3357 ATE 4–1 are shown anchored to the relevant repeat. The positions of the left and right repeats are depicted by the arrows in the inset (from S3H Fig) within each tree. The strains that are most distant to the left repeat tend to be closest evolutionarily to the right repeat, suggesting that each repeat served independently as an H class progenitor.

There are some similar features uniting the H class repeat with the TEs, however. Sequence alignment of H class repeats (S5B Fig) reveals that much of the variation in homology is due to preferential G>A and C>T transversion events (100% in NRRL 3357 ATE 6–1 and 80% in NRRL 3357 ATE 8–1). These biases are consistent with a RIP-like mutational process acting during the evolution of both TEs and H class repeats and hint at a common origin.

The remaining class repeats are the approximately 3.6 kb K and the 7.5 kb L repeats. Both are unrelated to subtelomeres, centromeres and any identifiable TEs. Both repeats are widely spread in *A. flavus* and *A. oryzae* strains and do not exhibit the polarity restrictions of the subtelomeric elements (S3K Fig). It therefore may represent a 'RIPed' copy of a hitherto undefined TE, although other possible explanations cannot be excluded.

## Discussion

In this study, I take advantage of a group of sequenced genomes from 14 strains of *Aspergillus flavus* to identify regions of genome instability. I characterized the predominant classes of large insertions representing either coding or non-coding elements. Non-coding sequences are composed of lengthy regions of 65–93% A+T-rich DNA. Phylogenetic analysis revealed that these AT insertions are dispersed to other chromosomal sites at higher frequencies than non-AT insertions. Despite the mobility and a higher mutation rate, AT insertions are nonetheless maintained and have a similar loss rate as non-AT insertions, consistent with a selection for their maintenance.

I conducted a global analysis of AT elements of $\geq$ 2 kb in three evolutionary-diverse strains: NRRL 3357, CA14, and SU-16. While the number of ATEs were similar in the strains, their length, distribution, and repetition varied significantly. The strains with complete telomere-to-telomere sequences, NRRL 3357 and CA14, showed a far-greater-than-expected frequency of these elements in regions within the subtelomeric 100 kb. The other major site of ATEs is the long ~100 kb that encompasses each centromere. I also identified ATEs either adjacent or closely positioned to the telomere (Class H) and in subtelomeric regions devoid of assayable TE homology (Classes A, F-J) which, unlike TEs, displayed polarities that were constant with respect to the telomere. The involvement of some of these ATEs in recombination is supported by the analysis of the end-specific dispersion of the class H repeats.

Classes B-F contain sequences homologous to transposons and retrotransposons. These elements were identified as evolutionary derivatives of TEs by the prevalence of transition over transversion mutations [58]. Many of these TE remnants have also undergone premeiotic repeat-induced point mutation. Previous studies have found that the percentage of the genome that is subject to RIP varies considerably from the extreme of 55% (in *Pyrenophora*) to moderate (13.3% in *Neurospora*) to relatively low levels (1.3% in *Aspergillus oryzae*) [29]. However, concrete evidence for *A. flavus* RIP has been challenging to determine, due both to the paucity of intact TEs in currently evolved strains and to the far less distinct AT-rich regions in *A. oryzae* (and *A. flavus*) than in *Neurospora* and some other fungi [59].

Since AT-rich regions may contribute significantly to genetic instability among divergent strains, I was interested in the contribution of RIP to the ATE intra-strain genetic variability. I therefore focused on the Gypsy 1, 2 and 4 intact elements to measure the relevance of RIP activity. The Gypsy class displayed a very high frequency of C>T mutations that may have proceeded close to substrate exhaustion, thereby retaining significant homology, presumably due to the slow rate of random mutation relative to RIP. The high density of sequences homologous to TEs supports multiple cycles of insertion, deactivation, and deletion.

One of the functions of ATE formation is the protection against the damaging effects of transposition through inactivation of these elements by RIP activity. However, my analysis suggests that these regions have evolved to perform new functions, at least at telomeres and centromeres. Several arguments support this view.

First, the similarity of ATEs between the centromeres of homologous chromosomes far exceeds centromeres of non-homologs in the three strains analyzed. Similarly, Gypsy elements in conserved centromeric clusters were nearly identical to one another but were more diverged when present in other regions of the centromere. The more ancient conservation of both homolog centromeric regions and TE homology likely points to a centromere-pairing-dependent role, such as meiotic recombination, centromere cohesion, or sister chromatid recombination [39], or to a role of the repeats themselves in the formation of a topologically competent centromere, as in higher eukaryotes. I note that the divergence of centromeric A+T-rich sequence in the three *A. flavus* strains analyzed may have deleterious consequences. Yeast studies have demonstrated that centromeres having imperfect homology, while still permitting meiotic separation, do so at lowered efficiency [60,61]. Such a decreased efficiency could ultimately lead to reproductive isolation between strains and may be a step in the process of speciation. The high level of homology between centromeres may protect against such isolation and help to maintain the ability of different strains to undergo a productive meiosis. These elements must therefore have evolved earlier from a common progenitor strain. Gypsy 1 and Mariner elements are also present at lower frequencies at the centromeres relative to non-centromeric locations. While the presence of conserved repeats in centromeres may play a part in centromere function, an excess of TE homologs may be deleterious by providing substrates for unequal sister chromatid exchange and inverted repeats, leading to dicentric or acentric chromosome formation. This may also be the factor in the selection for deleted forms of the TEs.

Second, a set of novel repeats is present at subtelomeres. Unlike TEs that display an overall random orientation, these repeats are oriented with a specific polarity with respect to the telomere and lack any discernable homology to TEs. I provided evidence for high levels of recombination at the evolutionary level at different telomere-adjacent regions, with sequence dispersion (and partial deletion) between homologous subtelomeres within and between strains. The preferential recombination between telomere-adjacent ATEs appears to have a bias towards specific chromosomal ends and is likely to reflect a telomere-specific function arising late after the evolutionary time of transposition and RIP. For example, the Class H element is conserved in telomere-adjacent sites of Chromosomes 4L, 8L, 6L, and 5L in NRRL 3357, CA14, and AF13 and to a less extent in SU-16, *A. oryzae* KBP3 and *A. sojae* SMF strains, where they nonetheless maintain their subtelomeric location (data not shown). Indeed, strain AF13 has a 4L subtelomere nearly identical to NRRL 3357. This repeat is therefore likely to be related to a progenitor element present in all strains, which subsequently recombines with a subset of ends in a strain-independent fashion.

Apart from a potential requirement for subtelomeric recombination activity, these subtelomeric elements may also act to disrupt telomere position effects (TPE). TPE is the silencing of telomere-adjacent genes that initiates at the telomere and spreads in a centromere-proximal direction through the formation of histone-modification-dependent heterochromatin [31].

TPE is present in at least some *Aspergillus* species, including *A. nidulans* (Palmer et al., 2010). However, conventional heterochromatin formation may be prohibitive in ATEs due to histone depletion that is predicted from the behavior of even short poly(dA): poly(dT) sequences in vivo [62,63]. The specifically modified histones required for telomere position effects are likely be absent in regions containing high A+T-DNA, raising the possibility that the subtelomeric ATEs may act to disrupt the spreading of TPE, thereby insulating coding regions. An additional possibility is that the recombination of ATEs at subtelomeres is involved in some aspect of telomere maintenance, such as found in the human ALT mechanism(s) of telomere elongation [64].

Third, studies using model substrates reveal that such histone-depleted chromatin architectures would be organized into large-scale chromosomal domains. Studies in the filamentous fungus *Epichloë festucae* have shown that such AT-rich domains establish organizing elements of gene transcription [65]. These regions include areas of repeated element inactivation, significantly affecting the three-dimensional topology of transcription units.

In summary, I propose an ATE cooption model, in which transposition and associated RIP gave rise to ATEs, which subsequently evolved novel cellular functions at telomeres and the centromeres of homologues (and possibly other sites). One of these functions may be homologous recombination, which may be promoted at telomeres, but be more restricted at the centromeres of non-homologous chromosomes. Future studies on the genetic manipulation of unique ATEs may uncover fundamental new directions in telomeric and centromeric function, as well as in gene regulation. I further anticipate that exploitation of the *A. flavus* strains will be an excellent evolutionary model to provide insights into the genetic instability of subtelomeric, centromeric, and gene cluster domains.

## Supporting information

**S1 Fig. Phylogenetic comparisons of a CA14 non-AT insertion with other *A. flavus* and *A. oryzae* strains.** Strains were compared by the Blast Tree Widget using the Fast Minimum Evolution algorithm, filtered at 90% homology. The tree is rooted in CA14 non-AT insertion 1–11. The scale for 0.009% homology difference is shown on the bottom right.
(PDF)

**S2 Fig. Phylogenetic Comparisons of a CA14 AT Indel with other *A. flavus* and *A. oryzae* strains.** Strains were compared as described in S1 Fig. The tree is rooted in CA14 AT insertion 1–4 (S5 Table). The scale for 0.01% homology difference is shown on the bottom right.
(PDF)

**S3 Fig. Classification of repeated ATE homologies into classes A-L.** A) Class A repeat organization measured by homology to NRRL ATEs 1–1 and 3–3; B) Class B repeat organization measured by homology to NRRL ATE 1–2; C) Class C repeat organization as assayed by homology to 3357 ATE 1–4; D); Class D repeat organization as assayed by homology to 3357 ATE 1–6; E) Class E repeat organization as assayed by homology to 3357 ATEs 1–6; F) Class F repeat organization as assayed by homology to 3357 ATE 1–3; G) Class G repeat organization as assayed by homology to CA14 ATE 3–3; H) Class H repeat organization as assayed by homology to 3357 ATE 4–1 and 6–1; I) Class I repeat organization as assayed by homology to 3357 ATE 3–1 and 7–1; J) Class J repeat organization as assayed by homology to 3357 ATE 8–2; K) Class K repeat organization as assayed by homology to SU-16 ATE 5–4. L) Class L repeat organization as assayed by homology to SU-16 ATE 8–1. Each class corresponds to the classes given by the superscripts in Table 2A, 2B, and 2C. In 3B-3E, green line, homology to the relevant ATE; the orange line, Gypsy elements; dark orange line, LTR 1 and LTR 2; red

line, Mariner elements; dark red line, Mariner 2N1 and Mariner 5; blue line, TAD 1 elements. The AO designation refers those TEs identified in Censor as *A. oryzae* derivatives. The AF designation is used for those Gypsy elements identified in *A. flavus* by BLAST querying for homologs to the Gypsy1 AO, Gypsy 2 AO, and Gypsy 4 AO elements. Some cryptic TEs were also identified by the homology search (e.g., the additional Mariner 1 elements in SU-16 ATE 5–3 and CA14 ATE 1–2). In Classes A and F-L, blue lines indicate homology to the indicated ATE. In the case of 3H, the number in parenthesis of the blue line indicates the position of homology within NRRL ATE 4–1. Relative directionality of homologous species in 3A and 3F-3L is indicated by the arrowhead.
(PDF)

**S4 Fig. Organization of Gypsy element homologs at chromosome 1, 6 and 8 centromeres in NRRL 3357, CA14 and SU-16.** The positions of the mutated Gypsy 2 and Gypsy 4 homologs within the centromeric domains of chromosome 6 (A), chromosome 8 (B) and chromosome 1 (C) (between the coordinates listed) were plotted and aligned among the three strains. The red, green, blue, and brown colors refer to regions of high homology as shown in Table 6A and 6B. The white color refers to regions of lower homology [corresponding to the black color in Table 6A and 6B)]. The GC content of these regions is presented above each chromosomal plot to aid in the identification of the centromere. Coding regions flanking the centromere are depicted by the purple arrows. Some chromosomes are inverted relative to left (L) and right (R) arms of NRRL 3357 chromosomes in the NCBI database, as indicated. Specifically, chromosomes 1 and 6 of CA14 are presented in inverted orientations. Chromosome 8 species are presented in inverted orientations in both CA14 and SU-16.
(PDF)

**S5 Fig.** Sequence analysis of representative H class repeats (a) and chromosome 4 centromeric Gypsy 4 elements (b). (A) BLAST search of NRRL 3357 ATE 1–4 (query) showing a portion of the homology with 3357 ATEs 6–1 (left) and 8–1 (right) (subjects). In this format, mismatches and gaps are represented as red residues. Quantification of the entire repeat in ATE 8–1 displayed 100% transition mutations, while ATE 6–1 displayed 80% transition mutations. All transversion mutations were A/T switches. (B) A portion of chromosome 4 centromeric Gypsy homolog (denoted by the green hue in Table 6 and Fig 3) homology identified in a BLAST search comparing strain NRRL 3357 (query) with strains SU-16 (A) and CA14 (B) (subjects), revealing a high level of sequence conservation.
(PDF)

**S1 Table. Positions and lengths of indels.** Positions of indels are presented using the coordinates of the inserted (top) strain in each strain a/strain b configuration. Red: AT-rich indel, Green: Reiterated AT-rich indel, White: non-AT indel, Blue: Reiterated non-AT indels. Other combinations of strains did not result in the identification of additional elements. Data, after conversion to the relative positions in NRRL 3357, are plotted in Fig 1.
(PDF)

**S2 Table. Summary analysis of 13 strains indicating the number of unique indels and the range of approximate insertion sizes.**
(PDF)

**S3 Table. Tabulation of AT Insertion in various chromosomes.** % AT content, approximate insertion position relative to NRRL 3357, and total fragment length, with the % of the insertion containing the AT-rich sequence in parentheses. CHR, chromosome number. When multiple

AT-rich regions are presented within the identified insert, they are separated by semi-colons.
(PDF)

**S4 Table. Summary of the total number of indels vs.** AT-indels on each chromosome in a comparison between CA14 and NRRL 3357 genomes.
(PDF)

**S5 Table. Tabulation of insertions in CA14 and NRRL 3357 genomes.** Data is derived from the indel analysis between the two strains, together with the chromosomal location, the approximate length of AT-rich regions, the site of the insertion, and the total length/AT element length % (parentheses) observed in the insertion. Consecutive numbers (separated by a semi-colon) indicate noncontiguous ATEs within the insertion.
(PDF)

**S6 Table. Mutations in Gypsy homologs are characteristic of RIP.** (left) The values of total mismatches/transition mutations (mm/Ts) between the AT-rich homologs and the Gypsy consensus sequence as determined by the Censor program are listed together with their standard deviation (SD) and sample size (n). (right) The fraction of the total transition mutations accounted for by G>A and C>T mutations within the consensus sequence was determined as described in Materials and Methods. Both full length and deleted forms of candidates were analyzed; nm, not measured.
(PDF)

## Acknowledgments

The author acknowledges Melody C. Baddoo and the Tulane Cancer Center Cancer Crusaders Next Generation Sequence Analysis Core for assistance with Next-Gen-Seq and bioinformatic analyses. I also thank Jeffrey Destruel, M.A. for technical assistance; Drs. Jeffrey Cary, Brain Mack, Mathew Lebar (Southern Regional ARS, USDA, New Orleans) and Dr. Astrid Engel for critical discussion of data; and Bonnie Hoffman for assistance with the preparation of this manuscript.

## Author Contributions

**Conceptualization:** Arthur J. Lustig.

**Data curation:** Arthur J. Lustig.

**Formal analysis:** Arthur J. Lustig.

**Funding acquisition:** Arthur J. Lustig.

**Investigation:** Arthur J. Lustig.

**Methodology:** Arthur J. Lustig.

**Project administration:** Arthur J. Lustig.

**Resources:** Arthur J. Lustig.

**Software:** Arthur J. Lustig.

**Supervision:** Arthur J. Lustig.

**Validation:** Arthur J. Lustig.

**Visualization:** Arthur J. Lustig.

**Writing – original draft:** Arthur J. Lustig.

**Writing – review & editing:** Arthur J. Lustig.

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
