## [Decision Letter · Decision Letter 0]

5 Jan 2023

PONE-D-22-32953Investigating the origin of subtelomeric and centromeric AT-rich elements in Aspergillus flavusPLOS ONE

Dear Dr. Lustig,

Thank you for submitting your manuscript to PLOS ONE. After careful consideration, we feel that it has merit but does not fully meet PLOS ONE’s publication criteria as it currently stands. Therefore, we invite you to submit a revised version of the manuscript that addresses the points raised during the review process.

A significant effort must be made so that the data presented is well described. The text and most tables need clarification in several places. Figures need to be of higher resolution for reviewers.Please address all specific points raised by the two reviewers.

We look forward to receiving your revised manuscript.

Kind regards,

Cecile Fairhead, Ph.D.

Academic Editor

PLOS ONE

Journal Requirements:

"This study was funded in part by USDA NCA 58-6054-0-014."

"AJL was funded in part by USDA NCA 58-6054-0-014. Arthur J. Lustig. The funders played no role in study design, data collection and analysis, decision to publish, or preparation of the manuscript.'

"I have no competing interest."

Reviewers' comments:

Reviewer's Responses to Questions

**Comments to the Author**

1. Is the manuscript technically sound, and do the data support the conclusions?

Reviewer #1: Yes

Reviewer #2: Yes

2. Has the statistical analysis been performed appropriately and rigorously? 

Reviewer #1: I Don't Know

Reviewer #2: N/A

3. Have the authors made all data underlying the findings in their manuscript fully available?

Reviewer #1: Yes

Reviewer #2: Yes

4. Is the manuscript presented in an intelligible fashion and written in standard English?

Reviewer #1: Yes

Reviewer #2: Yes

5. Review Comments to the Author

Reviewer #1: In the manuscript entitled “Investigating the origin of subtelomeric and centromeric AT-rich elements in Aspergillus flavus”, the author is looking at AT-rich regions in the genome of several strains and describe different classes depending on their localization, TE content and orientation.

This study is a thorough description of all these AT-rich elements and reveal interesting findings that raise some questions about the possible function of these elements.

The text is however a bit difficult to read and the poor quality of the figures and tables does not help.

First, the different classes of ATEs (from A to J) are not well defined if at all. There should be a clear definition of what characterizes each class in a short text or table. In addition, these classes should be associated with the three classes that are mentioned in the Abstract lines 25-26.

Other points to address (in order of text, regardless of importance):

- Line 66 : a reference about RIP should be added (Selker, 1987 for instance). Also, RIP is indeed pre-meiotic but there is no clear evidence that it happens during S phase.

- Line 80 : “silencer information regulators” is that a common name for proteins involved in position effect? If so, add a reference

- Line 92 : Change Plasmodia to Trypanosoma since the reference cited is only about T. brucei

- Line 107: “each strain having different size,” of what?

- Lines 110-111: “non-centromeric subtelomeric-enriched region” : unnecessarily complicated sentence: “subtelomeric” is enough

- Line 124-125: Where are the four strains from? Are the genomes published?

- Line 202: Rephrase. RIP mutates the C, the G->A is a consequence of that.

- Line 253, the title says “a major fraction” but line 255 it says 29%. Is 29% really a major fraction?

- Line 302: “First, related strains were most often located (in 20/24 cases) at allelic positions”, this must be rephrased, the strains are not located on the chromosome…

- Same thing line 304-305: “AT insertions had homologs on non-homologous chromosomes in different strains.”

- Line 321: “termed subtelomeres, indicated by the red font” where is the red font?

- Line 363: “Reanalysis of AT-element size in NRRL 3357 size ranged from 1.9 kb-21.2 kb.” Rephrase.

- Line 369: I don’t think that the right conclusion.

- Line 431: “RIP activity depends on the local context of the mutated cytosine” Do you mean in general or in Aspergillus? The mutation and methylation are not always context dependent.

- Line 427-448: Why didn’t you use something like RIPcal for such analysis?

- Line 566: flavus instead of flavis?

- Line 618: Neurospora should be italicized

Table 5 A: In the “Strain” column, “NRRL” is missing in front of 3357; also what does this column show? I guess it’s the strain from which the gypsy element is from. If so, why on line 9 the homology with NRRL3357 (itself) is not 100% (whereas it is with SU16)?

Table 5 A and B: in the legend: “(EP) positions noted in all three strains” I don’t see the coordinates for all 3 strains.

Supp table 3 : In columns 3 and 4: what does the fraction mean? E.g line 2, column 3: 4500/1700

Reviewer #2: This ms describes an interesting analysis of variable repeats in the genome of Aspergillus flavus, by comparing different strain sequences. Although the results are discussed in an interesting fashion, the general procedure followed, the classification of repeats, as well as the tables and figures, need clarification.

From the M&M, it seems that insertions, ie, variation between strains, were searched, and also AT rich regions in the most complete genomes, whether conserved or not, but it is unclear how the results from these searches were compared to one another and used.

Specific points that need to be adressed:

In the introduction, it is mentioned that the genome is compact: it would be good to remind its size and the number of protein-coding genes.

It would be good also to give the number of chromosomes and sizes. It is shown on figure 1, but never mentioned in the text.

In the results, subtelomeres are defined as 100kb regions at end, but surely this makes sense if all chromosomes are roughly the same size and several Mb large?

Table 1: even though the general idea comes across, the table is unclear.

First column has individual inserts from one strain, and following columns represent whole chromosome sequence from same strain and other strains, or?

What do the shaded and unshaded Xs represent in fact? Presence, absence, allelic or only non allelic?

In text describing table: “First, related strains were most often located (in 20/24 cases) at allelic positions in non-AT 303 insertions, when compared to either the NRRL 3357 or CA14. “

Please correct this sentence, as it is not the strains which are “located”.

Tables 2 A, B, C: The columns containing the information on Gypsy and Mariner is full of “0” and the information is in table 3, so maybe delete the columns in table2. “TAD” is supposedly “TAD1”?

Classification of ATEs: “The homologies fell into eleven classes (A-K).” It would be much clearer if a table describing the features of each class was provided.

In discussion, “While one of the functions of ATE formation is the protection against the damaging effects of transposition” is unclear. Does the author mean that the RIP mechanism which leads to ATEs , protects against transposition effects? If yes, please state this more clearly.

6. PLOS authors have the option to publish the peer review history of their article (what does this mean?). If published, this will include your full peer review and any attached files.

Reviewer #1: No

Reviewer #2: No

---

## [Author Response · Author response to Decision Letter 0]

16 Jan 2023

Response to Reviewers:

I would like to thank the editor and reviewers for their critique of the manuscript. I have attempted to address each issue as enumerated below. The page and line designations in the revised text refer to the marked up manuscript and are indicated in the bold font.

General Comment on Figure Resolution and Textual Clarity: 

Although the figures passed the PACE pre-flight analysis, I have increased the resolution of Figures 3-5 from 300 to 400 dpi. I have also increased the text size in Figures 3, 4, and 5 so that they are more legible when reduced in size.

Beyond the changes made to the text based on the critiques, the text has also been slightly modified for clarity in reading as indicated in the markup

Reviewer 1:

General Comments

1. I have added a new Table 3 (Line 461) that describes the class designations for Classes A-L (see note below) as requested. 

To correlate the Abstract more carefully with Results, I have modified the text (Lines 25-40) of the Abstract as requested so that the first two categories of ATE classes in non-centromeric regions are contiguous and specifically list classes A through J. [Classes K and L are not discussed in the Abstract. I have also eliminated the description of the term “classes” in the Abstract and Introduction as they related to types of ATEs with “categories” to eliminate confusion. 

I have also discussed the homolog classes in the section “Identification of multiple classes of homologous AT elements” and describe which classes are found in each strain as they relate to Tables 2A, 2B and 2C (Lines 389-398). 

[Note: The addition of Class L results from the two additional NRRL 3357 (ATE 8-5 and 8-6 in Table 2A) erroneously omitted from the original manuscript. The number of ATEs in NRRL 3357 has been adjusted accordingly in the text [Lines 341, 344, 345, 390]. Class L, found in three ATEs, is qualitatively like Class K (no identifiable TE element, not in subtelomeric regions, present in many A. flavus and A. oryzae strains) is discussed at the end of the Results sections (Lines 648-653). Class L has also been added to Supplementary Figure 3. 

Specific Comments:

1. Line 66: The Selker et al reference has been added to the RIP references. I have clarified that RIP occurs in pre-meiotic stages, rather than in a specific stage of meiosis. Line 74. Reference 26.

2. Line 80: Silencer information regulators (SIR) proteins are silencing- and heterochromatic-related proteins first identified in budding yeast (as referenced now) and subsequently in other organisms (Line 88 reference 34). 

3. Line 92: Plasmodia has been changed to Trypanosoma (Line 100).

4. Line 107. I now state that “each strain has a different ATE size…”(Line 115).

5. Line110: The phrase “non-centromeric” has been removed (Line 118-119)

6. Line 124: The four sequenced genomes (NRRL 2999), VCG1, VCG4, and A1) with their accession numbers and the published reference are now presented (Lines 133-135 reference 45). 

7. Line 202: I explain that RIP results from “a cytosine residue … mutating to a thymidine (and the corresponding guanine to adenine mutation on the complementary strand),”…(Lines 214-216).

8. Line 253: The Reviewer is correct that major (suggesting a majority) should be eliminated. I replaced it with “significant” (Line 265).

9. Line 302: “Strains” were replaced with “genomes”: “First, non-AT insertions from the genomes of related strains were most often located (in 20/24 cases) at allelic positions in non-AT insertions” (Line 323; also 305-320). 

Line 304, I have deleted “in different strains” (Line 326).

10. Line 321: The subtelomeric regions are indicated by the yellow shaded entries in the box (not by red font) as is now stated (Line 343).

11. Line 363: I have corrected the indicated sentence to state: “Reanalysis of AT-element size NRRL 3357 led to a revised size ranges from 1.9 kb-21.2 kb (Lines 390-391).

12. Line 369: The conclusion omits that differing ATE element size is a major difference between the genomes of the strains. Rather, I state “These data revealed that ATE length and chromosomal positions varied significantly among the different genomes.” (Lines 396-398).

13. Line 431: I did not intend to convey the impression that RIP is always context dependent. I now state: “RIP activity can depend in some fungi on the local context of the mutated cytosine residues and ultimately gives rise to AT-rich sequences” (Line 488)

14. Line 427: Both RIPcal and RIPper are great tools for genome analysis over longer domains they are limited in context of these studies. I state in the revised text:” …RIPper and RIPCAL [51, 57], two algorithms for measuring likely RIP frequencies over genomic distances, do not provide interpretable information for the relatively short and highly mutated sequences under analysis here” (Lines 482-485).

15. Line 566: “Flavis” has been corrected to “Flavi”, the name of the clade containing several species related to flavus (Line 630).

16. Line 618: Neurospora has now been italicized (Line 684). 

Table 5A, B (now Table 6):

1. We now state explicitly that the second column refers to the strain containing the Gypsy element used for comparison in the analysis (Lines 563-564, 574-575) We have corrected line 9 of Figure 6A to the appropriate NRRL 3357 standard of 100% . In addition, the term “NRRL” has been added as requested.

2. EP is noted only in the strain used for comparison. I have therefore deleted the phrase “in all three strains” (Lines 563, 573).

Supplementary Table 3

1. I had used the slash as an indication of two non-contiguous fragments in the AT-rich region as better stated in the Legend to Supplementary Table 5. However, the slash looks like a fraction and have therefore eliminated them from both Tables with an explanation that the semi-colon refers to non-contiguous regions (Lines 1050-1051, 1060).

Reviewer # 2:

General comments:

The original indel analysis provided the logic for the search for ATEs and the initial data for the dispersion of the ATEs. I have tried to specify the relationship more clearly between the indel and the ATE searches by noting each of the ATEs that were first found by the indel analysis in Tables 2A and 2B as marked by the hashtag (indicated in the Table legends) (Lines 361, 369-370). In addition, I have added a line in the text stating “that most of the insertions identified in the indel analysis between CA14 and NRRL 3357 (indicated by the hashtag in Tables 2A and 2B) were also found in these more detailed analyses (Lines 350-352). I have also supplied the coordinates for the insertions found from the indel analysis in Supplementary Table 3 so that they can be compared with the coordinates for the ATEs in Table 2.

Specific Comments:

1. In the revised text I have provided information regarding the total size of the genome, the range of sizes of chromosomes 1-8 and the estimated number of genes (13,500) in the first paragraph (Lines 43-47): “Aspergillus flavus is an opportunistic pathogen in plants and humans. Its 37.5 Mb genome is separated into 8 chromosomes ranging in size from 6.5 Mb of chromosome 1 to 3.25 Mb of chromosome 8. The organization of Aspergillus genomes is unusual. Like many fungi, the genome is relatively compact containing approximately 13,500 genes in its genome [1] (on average, 1 gene per 3 kb of DNA).” For simplicity, the subtelomeric regions are defined operationally as the terminal 100 kb at each of the chromosomes.

2. I agree that Table 1 is confusing as presented and have reconfigured both the Table and the Table Legend. The first column represents individual inserts from NRRL 3357, and the other columns represent the chromosome that contains a homolog in the genomes of both NRRL 3357 and/or other strains. I have now separated the previous yellow regions into two subsets. The Xs indicated by a yellow shading are insertions presented at the allelic site in the same chromosome in both NRRL 3357 and other strains. Those indicated by the blue shade are insertions at non-allelic sites at one or more positions within other chromosomes. Those X that are unshaded are insertions in strains other than NRRL 3357. The textual changes are described in Lines 305-3200.

3. The text describing Table 1 has been changed to state: “First, non-AT insertions from the genomes of related strains were most often located (in 20/24 cases)” (Line 323).

4. Table 2A, 2B, and 2C. I have eliminated the Table description of TEs within each ATE and pooled that data in an extended Table 4A (Line 467) (changed from Table 3A by the addition of a new Table 3). ”TAD” has been changed to “TAD1”

5. A clarification of the classes is provided by textually describing the classes found in each strain in Tables 2A, 2B, and 2C. Note the addition of a new Class L as indicated by the note to Reviewer 1. A summary of each class is also provided by a new Table, Table 3 (Line 461) that should enhance textual understanding.

6. In the Discussion, the manner in which ATEs may protect against transposition is now stated directly: “One of the functions of ATE formation is the protection against the damaging effects of transposition through inactivation of these elements by RIP activity. However, my analysis suggests that these regions have evolved to perform new functions, at least at telomeres and centromeres.” (Line 694-695).

---

## [Editor Report · Decision Letter 1]

30 Jan 2023

Investigating the origin of subtelomeric and centromeric AT-rich elements in Aspergillus flavus

PONE-D-22-32953R1

Dear Dr. Lustig,

We’re pleased to inform you that your manuscript has been judged scientifically suitable for publication and will be formally accepted for publication once it meets all outstanding technical requirements.

Kind regards,

Cecile Fairhead, Ph.D.

Academic Editor

PLOS ONE
---

## [Editor Report · Acceptance letter]

1 Feb 2023

PONE-D-22-32953R1 

Investigating the origin of subtelomeric and centromeric AT-rich elements in *Aspergillus flavus*

Dear Dr. Lustig:

I'm pleased to inform you that your manuscript has been deemed suitable for publication in PLOS ONE. Congratulations! Your manuscript is now with our production department. 

Kind regards, 

on behalf of

Pr Cecile Fairhead 

Academic Editor

PLOS ONE